# Improving the Hydrolysis Rate of the Renewable Poly(hexamethylene sebacate) Through Copolymerization of a Bis(pyrrolidone)-Based Dicarboxylic Acid

**DOI:** 10.3390/polym11101654

**Published:** 2019-10-11

**Authors:** Geert. J. Noordzij, Manta Roy, Natasja Bos, Vincent Reinartz, Carolus H.R.M. Wilsens

**Affiliations:** 1Chemelot InSciTe, Urmonderbaan 20F, NL-6167 RD Geleen, The Netherlands; geert.noordzij@maastrichtuniversity.nl; 2Aachen-Maastricht Institute of Biobased Materials (AMIBM), Faculty of Science and Engineering, Maastricht University, Brightlands Chemelot Campus, 6167 RD Geleen, The Netherlands; manta.roy@maastrichtuniversity.nl (M.R.); n.bos1@student.avans.nl (N.B.); vincent.reinartz@maastrichtuniversity.nl (V.R.)

**Keywords:** bis-pyrrolidone dicarboxylic acid, aliphatic polyester, hydrolysis, enzymatic depolymerization, itaconic acid, polycondensation

## Abstract

In this work, we report on the synthesis of a series of polyesters based on 1,6-hexanediol, sebacic acid, and *N*,*N*’-dimethylene-bis(pyrrolidone-4-carboxylic acid) (BP-C_2_), of which the latter is derived from renewable itaconic acid and 1,2-ethanediamine. Copolymers with a varying amount of BP-C_2_ as dicarboxylic acid are synthesized using a melt-polycondensation reaction with the aim of controlling the hydrolysis rate of the polymers in water or under bioactive conditions. We demonstrate that the introduction of BP-C_2_ in the polymer backbone does not limit the molecular weight build-up, as polymers with a weight average molecular weight close to 20 kg/mol and higher are obtained. Additionally, as the BP-C_2_ moiety is excluded from the crystal structure of poly(hexamethylene sebacate), the increase in BP-C_2_ concentration effectively results in a suppression in both melting temperature and crystallinity of the polymers. Overall, we demonstrate that the BP-C_2_ moiety enhances the polymer’s affinity to water, effectively improving the water uptake and rate of hydrolysis, both in demineralized water and in the presence of a protease from *Bacillus licheniformis*.

## 1. Introduction

The development of both renewable monomers and polymers has received considerable interest in recent years, resulting from both declining petroleum resources and increasing environmental awareness [1,2]. An end-of-life scenario, for example biological, chemical, or mechanical recycling should be envisioned when designing polymers that are to be used in a circular economy. In particular biological recycling, also known as biodegradation, could be of interest for polymers that are likely to end up in nature or other bioactive environments [3]. However, as is described by Swift [4], biodegradable polymers should undergo complete biodegradation in order to be considered biologically recyclable. In other words, the observance of progressive weight loss of polymers exposed to bioactive environments does not necessarily confirm their biodegradation; the polymer could erode into smaller (soluble) fragments that accumulate in nature without further degradation, as is generally the scenario for oxo-degradable polyolefin materials.

To this end, polyester materials are promising candidates for the design of biodegradable polymers as the presence of ester bonds in the backbone makes these materials susceptible to depolymerization by hydrolysis. In turn, full biodegradation is achieved when the hydrolyzed products can be mineralized by micro-organisms [4]. Indeed, hydrolysis and biodegradation of polyesters such as poly(ε-caprolactone), polylactides, polyhydroxyalkanoates, poly(1,4-butylene succinate), poly(hexamethylene sebacate), and other aliphatic polyesters has been reported [5,6,7,8,9]. For example, as is reported by Tokiwa and coworkers [9], poly(hexamethylene sebacate), having a melting temperature of 74 °C and a *M*_n_ of roughly 6 kg/mol, is readily hydrolyzed by *Rhizopus arrhizus* lipase, yielding the biodegradable 1,6-hexanediol [10] and sebacic acid [11] as products. Despite the fact that their chemical features favor biodegradability, other parameters such as the molecular weight, hydrophobic/hydrophilic properties, and crystallinity determine the depolymerization kinetics [5,12,13,14]. In particular, the presence of crystals and a rigid amorphous phase retards the depolymerization process in polyesters as the chains inside these rigid domains are less susceptible to hydrolysis [15]. Nevertheless, the presence of crystals in aliphatic polyesters is often required as this imparts high temperature stability and provides the desired mechanical performance.

An interesting renewable monomer that has proven useful for the synthesis of renewable polymers is itaconic acid [16,17]. In particular, the aza-Michael addition between the α,β-unsaturation of itaconic acid with an alkyl-amine can be used to generate renewable pyrrolidone-based dicarboxylic acid (BPDA) monomers. In turn, these BPDA monomers have successfully been used for the synthesis of polyesters [18,19], polyamides [20,21,22,23], and poly(ester-amide)s [24,25,26]. Though the presence of the cyclic pyrrolidone moieties are generally used to enhance the glass transition temperature [18], they also enhance the polymer’s affinity to water, resulting in an increased moisture uptake. In turn, the absorbed water acts as a plasticizer and facilitates a suppression in glass transition temperature [23,24]. Consequently, the affinity to water combined with the stereochemical complexity and overall bulkiness of the pyrrolidone ring generally prevents crystallization of BPDA-based polymers. Though often amorphous, polymers containing aliphatic BPDAs are reported to be bioresorbable, in particular when exposed to UV light or under alkaline conditions as this facilitates hydrolysis of the pyrrolidone ring. For example, the polymer of *N*,*N*’-dimethylene-bis(pyrrolidone-4-carboxylic acid) (BP-C_2_) and 1,6-hexanediol with a glass transition temperature of 24 °C and a *M*_w_ of 78 kg/mol is reported to hydrolyze fully upon immersion in water for one year (*M*_w_ of 688 g/mol) [18]. Given the biodegradability of the structurally similar n-methyl pyrrolidone (NMP), BPDAs might also be biodegraded by bacteria belonging to the *Pseudomonas*, *Paracoccus*, *Acinetobacter*, and *Rhodococcus* genera existing in sewage [27,28,29]. However, we would like to stress that as the biodegradability of these monomers is not confirmed, this remains a point for future investigations.

Nevertheless, these BPDA monomers could prove useful in controlling the hydrolysis rate of polyesters; the inability of this moiety to be incorporated in a polymer crystal will ensure its presence in the amorphous phase, thereby enhancing both the water uptake and hydrolysis of the amorphous phase without significantly affecting the crystalline domains. To evaluate this potential, in this work we report on the synthesis, characterization, and hydrolysis studies of copolymers of the bioresorbable (BP-C_2_, **1**), dimethyl sebacate (**2**), and 1,6-hexanediol (**3**), as shown in Scheme 1.

## 2. Materials and Methods

### 2.1. Materials

Itaconic acid, 1,2-ethanediamine, 1,6-hexanediol, dimethyl sebacate, and Ti(IV) *n*-butoxide were obtained from Sigma Aldrich. Standard laboratory solvents were obtained from Biosolve. Deuterated solvents were obtained from Buchem BV (Apeldoorn, The Netherlands). The purchased compounds were used directly without further purification, unless otherwise specified.

### 2.2. Synthesis of N,N’-Dimethylene-bis(pyrrolidone-4-carboxylic acid)

The synthesis of *N*,*N*’-dimethylene-bis(pyrrolidone-4-carboxylic acid), abbreviated as BP-C_2_, was performed using a protocol described in our earlier work and is briefly recalled [23]. Itaconic acid (13.01 g, 0.10 mol) and 1,2-ethylenediamine (3.25 ml, 0.05 mol) were heated in a round-bottom flask with a catalytic amount of water at 130 °C for 18 h. After this period, the residual water was removed using reduced pressure, after which a viscous liquid was obtained. The product was crystallized from a mixture of water and methanol, and dried in vacuum oven at 80 °C, to yield off-white crystals with a yield of 95%. Purity (≥99%) of the monomer was determined using NMR analysis. ^1^H NMR (CDCl_3_ + *d*-TFA, 300 MHz): δ 3.89 (m, 5.2 H, γ and δ‴ in Figure 1), 3.62 (m, 1.6 H, δ″ in Figure 1), 3.43 (m, 3.2 H, α and δ′ in Figure 1), 2.87 (m, 4 H, β in Figure 1). ^13^C NMR (CDCl_3_, + *d*-TFA 300 MHz): δ 177.3 (C=OOH), 176.6 (NC=O), 49.3 (NCH_2_ ring), 39.7 (NCH_2_ spacer), 35.6 (CH ring), 33.1 (CH_2_ ring).

### 2.3. Melt Polycondensation Procedures

Polymerization of dimethyl sebacate, 1,6-hexanediol, and BP-C_2_ was performed in a two-step melt-polycondensation reaction—an initial esterification (oligomerization) step, followed by transesterification (polymerization) in the melt under deep vacuum (<0.01 mbar). The ratio of 1,6-hexanediol to dicarboxylic acid was kept at 1.1:1, whereas the ratio of dimethyl sebacate and BP-C_2_ was varied to obtain copolymers with different amounts of BP-C_2_. The polymers were named such that they reflect the percentage of BP-C_2_ moieties of the overall dicarboxylic ester content. For example, the polymer containing 10% BP-C_2_ and 90% dimethyl sebacate is named **10% BP-C_2_**.

First, the monomers were loaded in the desired ratio in a round-bottom flask together with 0.5 wt% Ti(IV) *n*-butoxide and were heated to 180 °C under nitrogen flow. The mixture was maintained at 180 °C for 6 h after a homogenous mixture was obtained. Due to the high melting temperature of BP-C_2_, a few drops of DMF were added to the polymerization mixtures containing ≥25% BP-C_2_ as dicarboxylic acid to ensure a homogenous mixture. Note, the use of DMF is not desired as it is not an environmentally friendly solvent. Given that polymers containing less than 10% BP-C_2_ as dicarboxylic acid or less can be synthesized in the melt, the use of such low amounts of BP-C_2_ is preferred. After the oligomerization step, the reaction mixture was placed under reduced pressure for 4 to 6 h at 180 °C. Additionally, the polymers were polymerized at 200 °C for 2 h under reduced pressure to further increase the molecular weight. The polymers were isolated from the reactor flask after cooling to room temperature. An overview of the synthesized polymers, their molecular weights, and the BP-C_2_ content is provided in Table 1. The representative example of the synthesis of **10% BP-C_2_** is given; BP-C_2_ (0.57 g, 2.0 mmol), dimethyl sebacate (4.28 g, 18 mmol), and 1,6-hexanediol (3.17 g, 22 mmol) were loaded in a 100 mL 3-neck round-bottom flask, equipped with an N_2_/vacuum inlet, Vigreux column, and a heavy duty mechanical stirrer. The reaction mixture was purged 3 times with a vacuum/N_2_ cycle before slowly heating to 180 °C, prior to adding Ti(IV) *n*-butoxide (40 mg, 0.12 mmol). The reaction mixture was heated to, and maintained at, 180 °C for 6 h during which the generated methanol and water were distilled off. Next, reduced pressure was applied at 180 °C for 6 h. In a second step, the polymer was heated to 200 °C for two hours and under reduced pressure. The polymer was obtained as a yellow solid after cooling and was used without further purification.

### 2.4. Water Absorption and Hydrolysis Experiments

The synthesized polymers were cut into small square samples of 10 ± 1 mg (or rolled into small balls for the polymers having a *T*_g_ below room temperature) and were immersed in demineralized water. After immersion for one day, the polymers were removed from the water phase and their surface was dried carefully prior to TGA for determination of the amount of absorbed and adsorbed water. Similarly, samples were placed in demineralized water for one week, three weeks, and twelve weeks, after which they were removed from the water phase, dried in vacuo, and subjected to gel permeation chromatography (GPC) analysis. Additionally, after immersion of the polymers containing 0%, 50%, and 100% BP-C_2_ as dicarboxylic acid for 12 weeks in demineralized water, the water phase was analyzed using LC-MS to identify water-soluble fragments.

### 2.5. Enzymatic Depolymerization Experiments

The enzymatic hydrolysis of the polymers was facilitated using the following procedure. In general, the polymers were cut into squares or rolled into balls of approximately 20 mg. Next, the samples were immersed into PBS buffer (1 ml, 100 mM), together with a protease from *Bacillus licheniformis*, (Sigma Aldrich, Zwijndrecht, The Netherlands); 1–2 mg of the protease enzyme was added to the polymer together with 40 µl of CaCl_2_ solution (0.01 M) at 37 °C under constant shaking. After 24 h, the residual samples were removed from the solution, dried in vacuo overnight at 40 °C, after which their residual weight was determined.

### 2.6. Characterization Methods

NMR spectra were recorded with a Bruker Ultrashield 300 spectrometer (300 MHz magnetic field). Generally, NMR samples were prepared by dissolving ca. 10 mg of monomer or polymer in 0.5 mL deuterated chloroform and a few drops of deuterated trifluoroacetic acetic acid (CDCl_3_/*d*-TFA). All spectra were referenced against tetramethylsilane (TMS).

Molecular weight (*M*_n_, *M*_w_) and dispersity (Ð) of the polymers were determined from gel permeation chromatography (GPC), using chloroform as eluent. Analysis was performed with a Prominence-I LC-2030 equipped with a Shodex GPC KF-805L column. Analytical grade CHCl_3_ was used as the mobile phase at 40 °C, with a flowrate of 1 ml/min. GPC samples were prepared by dissolving ca. 3 mg of polymer in 1.5 ml of solvent overnight under constant shaking, and then the samples were filtered over a 0.2 μm PTFE syringe filter prior to injection. For the determination of the molecular weight of the polymers immersed in demineralized water, the polymers were removed from the water phase, dried, and dissolved in chloroform prior to measurement.

LC-MS analysis was performed using an LC-MS-2020 from Shimadzu equipped with a VisionHT C18 HL 1.5 µm column. A gradient of analytical grade ACN and MilliQ water with 0.1% formic acid was used a mobile phase at 30 °C, with a flowrate of 1 ml/min.

Attenuated total reflection Fourier transform infrared spectroscopy (ATR-FTIR) was performed using a PerkinElmer Spotlight 400 equipped with a PIKE GladiATR, dual mode MCT (mercury cadmium telluride) detector with an array or a temperature-stabilized DTGS (deuterated triglycine sulfate) as a standard configuration. Spectra were collected in the range 450–4000 cm^−1^ with a spectral resolution of 4 cm^−1^ at 100 °C, a temperature above the melting temperature of the synthesized polymers, which allows for interpretation of the backbone structure without overlapping vibrational bands of the moieties present in the crystalline domains.

The thermal stability and water absorption of polymers was determined through thermogravimetric analysis (TGA) using a TA Instruments Q500 Experiments were performed under a nitrogen atmosphere with a heating rate of 10 °C/min.

The thermal behavior of the synthesized polymers was determined via differential scanning calorimetry (DSC) using a TA Instruments DSC Q2000. Typically, two heating and cooling runs were performed at a rate of 10 °C/min; the first heating was used to erase any thermal history in the samples. Glass transition temperature (*T*_g_), peak melting temperature (*T*_m_), enthalpy of melting (Δ*H*_m_), peak crystallization temperature (*T*_c_), and the enthalpy of the crystallization exotherm (Δ*H*_c_) were obtained from the second heating and cooling run. DSC samples were prepared by loading 3–5 mg oven-dried samples in Tzero Hermetic Aluminum pans.

Wide angle X-ray diffraction (WAXD) analysis was performed on the synthesized polymers using a SAXSLAB Ganesha diffractometer, with a sample to detector distance of 80 mm using CuKα radiation (λ = 1.5406 Å). Crystallinity was calculated using the interactive peak fitter in MATLAB, by taking the ratio between area of the amorphous halo and the area of the two diffraction signals at 21 and 25 2θ. An example of the peak fitting results for the polymer **0% BP-C_2_** is provided in the Appendix A.

## 3. Results and Discussion

### 3.1. Polymer Synthesis

In the polymerization reactions performed in this study, as shown in Scheme 1, we made use of the BP-C_2_ in the dicarboxylic acid form, as it can be synthesized using a green route in high yield and purity. The synthesis was performed in bulk with water as catalyst, followed by purification through recrystallization in a mixture of water and methanol. As explained in the experimental section, the polymers were named such that they reflected the percentage of BP-C_2_ moieties of the overall dicarboxylic ester content. For example, the pure poly(hexamethylene sebacate), having no BP-C_2_, was named **0% BP-C_2_**. As is observed from Table 1, the molecular weight of the polymers does decrease slightly with increasing BP-C_2_ content, suggesting a lower reactivity of the BP-C_2_ monomer compared to dimethyl sebacate. Nevertheless, the use of BP-C_2_ does not appear to limit the molecular weight build-up during polymerization as we obtain polymers with a molecular weight (*M*_w_) close to 20 kg/mol or higher, values common for polymers synthesized via melt-polycondensation routes. However, the dispersity of the materials is relatively high, likely resulting from the absence of a purification step. As can be observed from the GPC traces discussed in Section 3.3 and provided in the Appendix A, the polymers contain some low molecular weight components corresponding to the presence of oligomers and/or cyclic species.

The presence of BP-C_2_ is detected from ^1^H NMR analysis, as shown in Figure 1, by the characteristic multiplet around 2.7–3.0 ppm, corresponding to protons next to the carbonyl of the pyrrolidone ring (resonance β), as shown in Figure 1 [23]. The BP-C_2_ percentage incorporated in the polymer was determined from the ratio between the integral of signal β and the integral of the resonance corresponding to the methylene protons next to the carbonyl signal of the sebacic acid moiety (resonance A), as shown in Figure 1 and Appendix A. Note, the signal for protons δ split up into three peaks as a result of the different rotational conformations of the pyrrolidone rings and the corresponding interactions of the protons δ with the carbonyl in the pyrrolidone rings [25]. The ^1^H NMR traces used for the determination of the BP-C_2_ percentage of the other polymers are provided in the Appendix A. Note, the location of resonance β is dependent on the used amount of deuterated trifluoroacetic acid (*d*-TFA) for dissolution of the polymers; in general, we observed a downfield shift of this resonance when using a higher concentration of *d*-TFA. This shift is expected to be the result of the acceptation of a proton by the pyrrolidone ring and the concomitant change in electronegativity.

Additionally, ATR-FTIR analysis was performed to visualize the presence of the BP-C_2_ moieties in the polymer backbone. As is depicted in Figure 2, we observed that the intensity of the carbonyl stretch (vibration **I**) of the ester bonds of polymer **0% BP-C_2_** has a higher intensity compared to polymer **100% BP-C_2_**. We expect that this is resulting from the decrease in ester bonds concentration in the polymer backbone, originating from the increased molecular weight of BP-C_2_ compared to the sebacate moieties (82 g/mol difference). Additionally, the introduction of an increasing amount of BP-C_2_ facilitates the rise and increase of vibrations **II** and **III**, which correspond to the pyrrolidone carbonyl stretch vibration and bend vibration of the methylene groups next to the BP-C_2_ amines [30], respectively. Indeed, we observed a linear increase in peak intensity of vibration **III** with the incorporated BP-C_2_ concentration as calculated from NMR, as shown in the inset in Figure 2. Overall, this indicates that the FTIR and NMR data are in good agreement and that we have successfully incorporated BP-C_2_ in the polymer backbone.

### 3.2. Thermal Behavior and Crystallinity

To evaluate the effect of the presence of BP-C_2_ in the polymer backbone on the thermal properties, DSC and TGA were performed on the synthesized materials, as shown in Table 2 and Figure 3. Generally, we observed that all polymers were stable and did not exhibit significant weight loss below 300 °C, as is indicated by the temperature where 10 wt % of the material was lost during TGA experiments (*T*_90_). This suggests that introduction of BP-C_2_ moieties in the polymer backbone does not have a negative effect on the thermal stability of the synthesized polymers. However, when looking carefully at Figure 3A, one can observe that the polymers with a low amount of BP-C_2_ display a slight weight loss above 250 °C, which we attribute to evaporation of the low molecular weight species observed in GPC analysis, as discussed in Section 3.3, and responsible for the broad dispersity of these samples in particular, as shown in Table 1.

Indeed, as expected, the incorporation of BP-C_2_ linearly increases the glass transition temperature from the literature value of –60 °C for polymer **0% BP-C_2_** [31] up to 5 °C for polymer **100% BP-C_2_**. Note, for the polymers **0% BP-C_2_**, **5% BP-C_2_,** and **10% BP-C_2_**, no *T*_g_ was observed during DSC analysis, likely resulting from the extensive crystallization during the cooling and the resulting low amorphous fraction. Interestingly, the *T*_g_ value for the polymer **100% BP-C_2_** was significantly lower than the 24 °C reported by Miller and coworkers [16] for the same polymer. We expect that this is resulting from the presence of the oligomeric and/or cyclic species in our polymers, effectively acting as a plasticizer for the polymer. Indeed, immersion of the polymer in demineralized water for one week results in slow diffusion of (at least part of) the low molecular weight species into the water phase, as discussed in Section 3.3, effectively increasing the observed molecular weight and the *T*_g_ to 15 °C (after drying), as shown by the green curve in Figure 4.

As mentioned earlier, the bulky BP-C_2_ moiety is not expected to co-crystallize in the monoclinic crystal phase of poly(hexamethylene sebacate) [32]. As can be observed from Figure 3B, both the polymer’s melting temperature (*T*_m_) and the melting enthalpy (Δ*H*_m_) decrease with the increasing amount of BP-C_2_. Such decreasing *T*_m_ and Δ*H*_m_ confirm that the BP-C_2_ moieties limit the crystallization of the poly(hexamethylene sebacate) segments, resulting in the formation of defected crystals with decreasing lamellar thickness. Interestingly, **25% BP-C_2_** has a double melting endotherm upon heating. Though the exact origin of this double melting transition is unknown, we speculate that this behavior originates from the presence of the defected crystals with a variation in melting temperature (and lamellar thickness, as mentioned before).

Although the polymer **50% BP-C_2_** does not show appreciable crystallization in DSC analysis while employing a cooling rate of 10 °C, slow crystallization proceeds at room temperature as is evident from the sample transforming from transparent to opaque during storage at room temperature, and from WAXD analysis, as shown in Figure 5. No crystallization is observed anymore upon the incorporation of more than 50% BP-C_2_ and amorphous materials are obtained instead.

As expected, the introduction of increasing amounts of BP-C_2_ decreases the overall crystallinity of the samples, as shown in Table 2, but does not affect the crystal structure, as shown in Figure 5. These findings confirm that the BP-C_2_ moieties do not participate in the crystallization and thus reside in the amorphous phase. Taking into account that the formed crystals solely contain sebacic acid as dicarboxylic acid, one can calculate the percentage of BP-C_2_ in the amorphous phase, as shown in Figure 5B. Indeed, we observed that exclusion of the BP-C_2_ moieties from the poly(hexamethylene sebacate) crystal structure resulted in an increased BP-C_2_ concentration in the amorphous phase compared to the average amount present in the polymer backbone (red dotted line). For example, we observed that the BP-C2 moieties in the **5% BP-C_2_** and **10% BP-C_2_** polymers were excluded from the crystalline domains, resulting in an approximate amount of 10% and 20% BP-C_2_ in the amorphous regions, respectively. Note, these values are obtained as follows. The generated crystals were presumed to solely consist of sebacic acid as the dicarboxylic acid component, hence, the fraction of poly(hexamethylene sebacate) in the amorphous phase can be approximated by subtracting the fraction of sebacic acid in the crystalline domains (indicated by the crystallinity) from the total amount of sebacic acid in the polymer. Next, the BP-C_2_ concentration in the amorphous phase can be calculated by taking the ratio between the sebacic acid in the amorphous phase and the BP-C_2_ concentration in the polymer backbone (which is presumed be solely present in the amorphous phase).

### 3.3. Water Absorption and (Enzymatic) Hydrolysis

The effect of the presence of BP-C_2_ moieties in the amorphous phase on the water uptake and (enzymatic) hydrolysis rate has been investigated using TGA, GPC, and HPLC analysis. Firstly, the synthesized polymers were immersed in water for one day to ensure maximum water absorption. Next, the samples were removed from the water phase, dried carefully with a cloth, and placed for TGA. During heating, we observed a gradual weight loss between 50 and 100 °C, corresponding to evaporation of the absorbed water, as shown in Table 2 and Appendix A. For polymer **0% BP-C_2_**, we observed a weight loss of only 0.3 wt%, likely corresponding to evaporation of adsorbed water on the surface. Such behavior is expected, given that the polymer is rather hydrophobic in nature. In contrast, for the polymers containing 10% and 25% BP-C_2_, the water uptake increases to 1.6 wt% and 4.8 wt%, respectively. When using even higher concentrations of BP-C_2_, the absorption of water increases significantly to values above 12 wt%. Overall, this data confirms that the copolymerization of BP-C_2_ can effectively be used to enhance the water uptake of polyesters.

Additionally, the polymers were immersed in demineralized water for one week, 3 weeks, and 12 weeks, after which they were removed from the water phase, dried in vacuo, and analyzed using GPC analysis. The characteristic GPC traces for the polymers containing 0%, 50%, and 100% BP-C_2_ are provided in Figure 6. The GPC traces of the other polymers obtained directly after synthesis and after 12 weeks of immersion in demineralized water are provided in Appendix A. As is observed from Figure 6, no significant change in the high molecular weight regime was observed for the polymers immersed in water for one or three weeks. However, we did observe a gradual decrease in the low molecular weight components, suggesting their slow dissolution and migration into the water phase. Additionally, when comparing the effect of hydrolysis on the molecular weight of the polymers after immersion in water for 12 weeks, we clearly observed that *M*_w_ decreases faster for polymers containing increasing amounts of BP-C_2_, as shown in Figure 6D. Indeed, when performing LC-MS analysis on the water phase, as shown in Appendix A, we observed the presence of the various monomers used in this study, in addition to some dimeric and cyclic species. More importantly, the concentration of dissolved monomers and oligomers increase significantly for polymers having an increasing amount of BP-C_2_, undoubtedly resulting from the faster hydrolysis and higher affinity/solubility of the BP-C_2_ with water. Overall, these findings confirm that both the water uptake, solubility, and the rate of hydrolysis is increased in polymers having an increasing BP-C_2_ concentration in the polymer backbone.

Lastly, we evaluated the effect of BP-C_2_ in the polymer backbone on the enzymatic hydrolysis. Generally, 20 mg polymer was placed in a vial together with 1 mL PBS buffer, 40 µL of 0.01 M CaCl_2_, and 1–2 mg of a protease enzyme from *Bacillus licheniformis*. The vials were closed and shaken at 37 °C for 24 h, after which the residual polymer was removed, dried, weighed, and analyzed via GPC. In general, we observed that both the amorphous polymers, **75% BP-C_2_** and **100% BP-C_2_**, significantly decreased in both size and weight after the enzymatic hydrolysis experiment. The enzymatic degradation generally occurs at the surface of the sample, after which the low molecular weight products dissolve, allowing other parts of the sample to be hydrolyzed [4]. Indeed, we observed that the residual weight decreased with increasing BP-C_2_ concentration in the polymer backbone, as shown in the red squares in Figure 7, undoubtedly resulting from the increased water absorption and the increasing amorphous fraction in the polymers, easing the accessibility of the ester bonds for the enzyme. Interestingly, when looking at the molecular weight of the polymers after enzymatic hydrolysis, as shown in the black circles in Figure 7, a different trend is observed; the molecular weight of the residual polymer **75% BP-C_2_** and **100% BP-C_2_** after enzymatic hydrolysis were not significantly lower compared to the molecular weight of the as-synthesized samples. This indeed confirms that the enzymatic hydrolysis proceeds in a ‘layered fashion’. Interestingly, the polymer **50% BP-C_2_** displays a significant decrease in molecular weight, combined with a weight loss that approaches 30%. Visually, we observed that the sample did not decrease in size like **75% BP-C_2_** and **100% BP-C_2_**, but instead fragments into smaller pieces. We consider it likely that the crystalline fraction maintains the structural integrity of the sample—though hydrolysis proceeds in the amorphous phase, segments of chains were anchored as they were part of a crystal, preventing their dissolution in water and allowing us to measure them during GPC analysis. Similarly, we expect that this also happens in the polymers having 25% BP-C_2_ and less, though the rate of hydrolysis will be even slower due to the decreased BP-C_2_ concentration and the increased crystallinity. Nevertheless, these preliminary experiments clearly demonstrate that the copolymerization of BPDA-based monomers is a viable route to enhance the hydrolysis rate of poly(hexamethylene sebacate), both in demineralized water and under bioactive conditions.

## 4. Conclusions

In this work, we reported on the successful synthesis and characterization of a series of copolymers based on 1,6-hexanediol, dimethyl sebacate, and BP-C_2_. The used polycondensation route allows for the development of polymers with molecular weights (*M*_w_) around 20 kg/mol and higher according to GPC analysis. Though poly(hexamethylene sebacate) is semi-crystalline, the BP-C_2_ moiety does not fit in its monoclinic crystal structure, resulting in a suppression of both the melting temperature and crystallinity for polymers containing an increasing BP-C_2_ concentration. Upon the usage of more than 50% BP-C_2_ as dicarboxylic acid, the materials become fully amorphous. The presence of the BP-C_2_ moieties enhances the polymer’s affinity with water, increasing the water uptake, and thereby the rate of hydrolysis and solubility of the low molecular weight components. The polymer containing solely BP-C_2_ as dicarboxylic acid is likely to find any use for structural applications given the absence of crystallinity and its low *T*_g_. However, the results in this work demonstrate that the introduction of the hydrophilic BP-C_2_ moiety as a co-monomer is a viable route to enhance the rate of hydrolysis of hydrophobic polyesters in the presence of water and under bioactive conditions, though only for the amorphous component of the polymers.

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
