# Peer review of "Improving the Hydrolysis Rate of the Renewable Poly(hexamethylene sebacate) Through Copolymerization of a Bis(pyrrolidone)-Based Dicarboxylic Acid"

_polymers, 2019, doi:10.3390/polym11101654_

Round 1
Reviewer 1 Report
In this manuscript, the authors report the synthesis of biodegradable renewable copolyesters from hexanediol, dimethyl sebacate and itaconic acid-based dicarboxylic compound (BP-C2).
The results and discussion are written well and there are a lot of valuable information in the manuscript. I think that the manuscript can be accepted after minor revision.
I would like to suggest a few comments:
Please check the description of the monomer ratio for dicarboxylic acids. I think the ratio is usually described as a molar ratio or molar percentage. In Table 1, however, the authors show the BP-C2 amount in weight percentage. Is it correct?
Usually, in an FT-IR spectrum, lower wavenumber region is shown at the right side. Please use reversed axis for the x axis (Figure 2).
Please check the description of the legend of Table 2. There may be an error.
Please give more account about how to calculate the BP-C2 amount in the amorphous phase (Lines 280-282, in page 7 and Figure 5 (B)).
I think that the degradation process by enzyme is affected by the substrate selectivity of the enzyme. How about discussing the substrate selectivity of the enzyme on the results in Figure 7?
I think that the mechanical properties of the copolyester containing larger amount of BP-C2 at room temperature is very low, because it has no crystal phase and the glass transition temperature is lower than room temperature. What do the authors think about any application of this polymer?
Author Response
Before providing point-by-point answers to the queries raised by the reviewers, we would like to express our gratitude for their efforts in taking time to evaluate our work and provide critical comments with the aim to improve the quality of this work.
Reviewer 1
In this manuscript, the authors report the synthesis of biodegradable renewable copolyesters from hexanediol, dimethyl sebacate and itaconic acid-based dicarboxylic compound (BP-C2). The results and discussion are written well and there are a lot of valuable information in the manuscript. I think that the manuscript can be accepted after minor revision.
I would like to suggest a few comments:
Please check the description of the monomer ratio for dicarboxylic acids. I think the ratio is usually described as a molar ratio or molar percentage. In Table 1, however, the authors show the BP-C2 amount in weight percentage. Is it correct?
Answer: Indeed, the reviewer is correct in pointing out this. The calculated and weighed amounts are based on molar percentages. This is described in the example provided in the experimental section, though it is not clearly emphasized in Table 1. Accordingly, we have denoted the amounts as mol% instead of %.
Usually, in an FT-IR spectrum, lower wavenumber region is shown at the right side. Please use reversed axis for the x axis (Figure 2).
Answer: The suggestion has been taken up and Figure 2 has been updated accordingly.
Please check the description of the legend of Table 2. There may be an error.
Answer: Indeed, the error has been rectified and the duplication of TGA has been removed.
Please give more account about how to calculate the BP-C2 amount in the amorphous phase (Lines 280-282, in page 7 and Figure 5 (B)).
Answer: The following section has been added the provide more detail on the determination of the BP-C2 amount in the amorphous phase:
“For example, we observe that the BP-C2 moieties in the 5% BP-C2 and 10% BP-C2 polymers are excluded from the crystalline domains, resulting in an approximate amount of 10% and 20% BP-C2 in the amorphous regions, respectively. Note, these values are obtained as follows: The generated crystals are presumed to solely consist of sebacic acid as dicarboxylic acid component, hence, the fraction of poly(hexamethylene sebacate) in the amorphous phase can be approximated by subtracting the fraction of sebacic acid in the crystalline domains (approximated by the crystallinity) from the total amount of sebacic acid in the polymer. Next, the BP-C2 concentration in the amorphous phase can be calculated by taking the ratio between the sebacic acid in the amorphous phase and the BP‑C2 concentration in the polymer backbone (which is presumed be solely present in the amorphous phase).”
I think that the degradation process by enzyme is affected by the substrate selectivity of the enzyme. How about discussing the substrate selectivity of the enzyme on the results in Figure 7?
Answer: Indeed, it is very likely that the BP-C2 and the sebacic acid substrates have different interaction with the enzyme. However, there are several other parameters that influence the rate of hydrolysis, which are crystallinity and the water absorption of the amorphous phase, as mentioned in the paper. Both these parameters govern the accessibility of the polymer chain by the enzyme, and we consider this at this stage the most critical parameter for the observed hydrolysis behavior. Instead, we would recommend to study the substrate selectivity on polymers fully dissolved in water, thereby eliminating the effects of crystallinity and water uptake. However, this would require the synthesis of a new series of different copolymers and we consider this to be outside the scope of this work.
I think that the mechanical properties of the copolyester containing larger amount of BP-C2 at room temperature is very low, because it has no crystal phase and the glass transition temperature is lower than room temperature. What do the authors think about any application of this polymer?
Answer: The mechanical properties of the amorphous polyesters are not considered to be usefull for application where structural integrity at room temperature is required. The reviewer is correct in his/her statement as the materials absorb water, have a glass transition temperature below room temperature and can thus be considered polymer melts at these temperature without any notable mechanical performance. Their handling is challenging and therefore we do not foresee much use for these materials. Instead, we consider the materials containing low amounts of BP-C2 more interesting as they can be used as alternatives to the poly(hexamethylene sebacate) with controlled crystallinity and hydrolysis rate. To stress this we have added the following section to the conclusions:
“The polymer containing solely BP-C2 as dicarboxylic acid is likely to find any use for structural applications given the absence of crystallinity and its low Tg. However, the results in this work demonstrate that the introduction of the hydrophilic BP-C2 moiety a co-monomer is a viable route to enhance the rate of hydrolysis of hydrophobic polyesters in the presence of water and under bioactive conditions, though only of the amorphous component of the polymers.“

Reviewer 2 Report
The present work describes the synthesis of a series of polyesters derived from 1,6-hexanediol and different mixtures of sebacic acid and a dicarboxylic acid (BP-C2) based on bispyrrolidone and specifically prepared from the renewable itaconic acid . In general, the work merits publication after considering different points:
The introduction section should give information of the previously published parent homopolymers (i.e. 0% BP-C2 and 100% BP-C2), including molecular weights, thermal properties and degradation data if available. Samples were studied as synthesized without further purification. This is in my opinion the most problematic point since usually a reprecipitation step is required in order to avoid the presence of oligomers. Authors are conscious of this issue through GPC data, the high polydispersity index and even TGA analyses. I suggest to purify at least one copolymer composition in order to corroborate the formulated hypothesis and also to have data from a copolymer without non representative small fractions. Page 3, line 95: Please, indicate the assignment of signals observed in the proton spectrum. Please, explain the d', d’’ and d’’’ signals (Figure 1). Table 1: The molecular weight seems to significantly decrease for high BP-C2 contents (e.g. 75% and 100%). A comment seems necessary. Page 6, FTIR analysis: Are the spectra normalized in order to deduce information from the intensity of band I? Table 2: It should be interesting to emphasize that DSC data correspond to samples cooled from the melt while the degree of crystallinity (X-ray analysis) correspond to the sample obtained directly from synthesis. In this way, the data for 50% BP-C2 will be more understandable (i.e. crystallinity of 8% and an endothermic melting peak with a very low enthalpy). Figure 3: a double melting peak is observed for 0% BP-C2 and 25% BP-C2 samples. Some comments are necessary. Water absorption and degradation: Samples with a similar and defined shape should be studied for comparative purposes. This shape information is missing. Can contact angle measurements be provided in order to evaluate the hydrophilicity of the different copolymers? Enzymatic degradation is usually related to a surface process. In this way, changes on molecular weight should be scarce as observed by the authors. Data for the 50% BP-C2 sample seems problematic. Have the authors studied the degradation of this sample in absence of the enzyme? The hydrolytic degradation may be a bulk process that influences significantly the molecular weight. Scanning electron micrographs seem necessary to evaluate the surface erosion process during degradation. Revise and remove, please: “Error! Reference source not found” that appear in the text each time that a Figure is mentioned.
Author Response
Before providing point-by-point answers to the queries raised by the reviewers, we would like to express our gratitude for their efforts in taking time to evaluate our work and provide critical comments with the aim to improve the quality of this work. Note, a document including figures mentioned in the text have been added as attachment.
Reviewer 2
The introduction section should give information of the previously published parent homopolymers (i.e. 0% BP-C2 and 100% BP-C2), including molecular weights, thermal properties and degradation data if available.
Answer: The requested information is provided in the introduction (references present in manuscript):
“Indeed, hydrolysis and biodegradation of polyesters such as poly(ε‑caprolactone), polylactides, polyhydroxyalkanoates, poly(1,4-butylene succinate), poly(hexamethylene sebacate) and other aliphatic polyesters has been reported. For example, as is reported by Tokiwa and coworkers, poly(hexamethylene sebacate) with Mn of roughly 6 kg/mol is readily hydrolyzed by Rhizopus arrhizus Lipase, yielding the biodegradable 1,6-hexanediol and sebacic acid as products.”
And
“For example, the polymer of N,N’-dimethylene-bis(pyrrolidone-4-carboxylic acid) (BP-C2) and 1,6-hexanediol with a glass transition temperature of 24 °C and a Mw of 78 kg/mol is reported to hydrolyze fully upon immersion in water for one year (Mw of 688 g/mol)”
Samples were studied as synthesized without further purification. This is in my opinion the most problematic point since usually a reprecipitation step is required in order to avoid the presence of oligomers. Authors are conscious of this issue through GPC data, the high polydispersity index and even TGA analyses. I suggest to purify at least one copolymer composition in order to corroborate the formulated hypothesis and also to have data from a copolymer without non representative small fractions.
Answer: Generally, we prefer to work with the polymer materials in the ‘as-synthesized’ form as this provides valuable information on potential side reactions and the formation of cyclic structures, which are inevitably formed during such polymerization reactions. More importantly, residual monomers, cyclics or oligomers are widely acknowledged to facilitate such a plasticizing effect, in addition to the presence of solvents and water. Furthermore, the materials synthesized in reference 18 were also used as synthesized, thus they are also likely to contain such cyclic structures and residual monomers. In other words, we expect that the differences in glass transition can solely be attributed to the differences in molecular weight (Note, the Mw in ref 16 is 78 kg/mol under optimized conditions, compared to the 18 kg/mol reported in our work).
That being said, we have attempted to precipitate the 100% BP-C2 polymer in ether, though we did not manage to isolate a fraction with a higher molecular weight than obtained after immersion in water (and shown in Figure 4 of the manuscript). Nevertheless, the provided data shows a clear trend in the Figure 1 below (trend line manually added to guide the eye), which at least makes the suppression in Tg consistent throughout the series:
Figure 1: Relation between measured Tg of the copolymers and their BP-C2 content.
Page 3, line 95: Please, indicate the assignment of signals observed in the proton spectrum.
Answer: The requested references have been made in the experimental section.
Please, explain the d', d’’ and d’’’ signals (Figure 1).
Answer: Signals d’, d” and d’’’ split up as a result of the different interactions of the protons with the carbonyls of the pyrrolidone rings. To clarify this, we have added the following sentence including a reference to previous work where this is discussed.
“Note, the signal for protons δ split up into three peaks as a result of the different rotational conformations of the pyrrolidone rings and the corresponding interactions of the protons δ with the carbonyl in the pyrrolidone rings.”
Table 1: The molecular weight seems to significantly decrease for high BP-C2 contents (e.g. 75% and 100%). A comment seems necessary.
Answer: Indeed, it is likely that the reactivity of the BP-C2 is lower than that of the dimethyl sebacate explaining the slightly decreasing molecular weight of the polymers with increasing BP-C2 content. Therefore, the following line has been introduced in the manuscript:
“As is observed from Table 1, the molecular weight of the polymers does decrease slightly with increasing BP-C2 content, suggesting a lower reactivity of the BP-C2 monomer compared to dimethyl sebacate.”
Page 6, FTIR analysis: Are the spectra normalized in order to deduce information from the intensity of band I?
Answer: No normalizations have been performed, these spectra displayed as collected. However, to obtain these spectra reflecting solely the composition of the polymers and not the effects of crystallinity we collected the FTIR spectra at 100 °C rather than at room temperature, as explained in the experimental section.
Table 2: It should be interesting to emphasize that DSC data correspond to samples cooled from the melt while the degree of crystallinity (X-ray analysis) correspond to the sample obtained directly from synthesis. In this way, the data for 50% BP-C2 will be more understandable (i.e. crystallinity of 8% and an endothermic melting peak with a very low enthalpy).
Answer: Indeed, this is a very good suggestion that makes the data more easily interpretable. The suggested explanation has been added as a footnote in Table 2 and reads:
“cCrystallinity was determined on the as obtained samples without further heat treatment.”
Figure 3: a double melting peak is observed for 0% BP-C2 and 25% BP-C2 samples. Some comments are necessary.
Answer: Indeed, in particular the sample containing 25% BP-C2 displays a prominent double meting behavior, which we consider the result of the presence of various defected crystallites that inevitably have different melting temperatures. Though normally we observe a rather broad melting endotherm upon heating in the presence of such defected crystallites, it appears that the 25% BP-C2 polymer apparently favors the formation of crystals having two relatively distinct melting temperatures. From WAXD analysis we can clearly observe that the crystal structure of the crystal remains the same in the presence of BP-C2, hence the origin of this behavior is likely to be attributed to differences in crystal size. However, as we currently have no means to verify this we have added the following sentence to the manuscript:
“Interestingly, 25% BP-C2 displays a double melting endotherm upon heating. Though the exact origin of this double melting transition is unknown, we speculate that this behavior originates from the presence of the defected crystals with a variation in melting temperature (and lamellar thickness, as mentioned before).”
Water absorption and degradation: Samples with a similar and defined shape should be studied for comparative purposes. This shape information is missing.
Answer: Indeed, the shape information was missing. Generally, we cut the samples with a razorblade into small squares/cubes of approximately 10 (hydrolysis) or 20 (enzymatic degradation) mg in weight. However, this proved challenging for polymers with 50% BP-C2 and more as these materials were soft and easily deformed. Instead, these samples were rolled into small balls, although these slowly lost their shape upon immersion in water due to water uptake and sample relaxation. The descriptions have been taken up in the manuscript.
Can contact angle measurements be provided in order to evaluate the hydrophilicity of the different copolymers?
Answer: Unfortunately, the handling of the samples is rather challenging, in particular for these amorphous in nature or having a very low crystallinity. These samples can be considered as polymer melts at room temperature and tend to absorb water upon storage, hence they tend to flow at room temperature. For this reason, we did not perform any further analysis on the current set of materials. Instead, as also suggested by reviewer 4, we aim to synthesize a series of copolymers with BP-C2 contents varying from 25 – 50 wt% BP-C2 and study these more in depth both for their enzymatic degradation behavior and their physicochemical properties.
Enzymatic degradation is usually related to a surface process. In this way, changes on molecular weight should be scarce as observed by the authors. Data for the 50% BP-C2 sample seems problematic. Have the authors studied the degradation of this sample in absence of the enzyme? The hydrolytic degradation may be a bulk process that influences significantly the molecular weight. Scanning electron micrographs seem necessary to evaluate the surface erosion process during degradation.
Answer: Indeed, this result is rather unexpected as it appears that the material hydrolyzes readily in bulk rather than the occurrence of only surface erosion. As suggested by the reviewer, we have performed SEM analysis on the surface of this sample after 1 day of immersion in PBS buffer (Figure below, left) and 1 day of immersion in enzymatic (Figure below, right). In general, we observe that the sample that was degraded enzymatically broke up in several smaller particles with a rather rough surface (right image), suggesting that is did swell significantly. However, as a result of the drying process, the sample shrinks which could explain the rough surface. Similar features are observed for the polymer immersed in only PBS buffer (left image), though not as significant as when immersed in a PBS buffer with enzyme. More importantly, the sample did not break down into smaller pieces. We consider it possible that the relatively large decrease in molecular weight results from the increased surface of the sample and thus an improved water penetration through the sample. (This would be in addition to the surface erosion) However, further study would be required to make any reliable statements on this topic, which we aim to take up in future work.
Figure 2. SEM surface morphology of the 50% BP-C2 polymer after 1 day immersion in PBS buffer (left) and in PBS buffer with enzyme (right). Note, the samples were dried in vacuo at 20 °C prior to the application of a thin gold coating.
Revise and remove, please: “Error! Reference source not found” that appear in the text each time that a Figure is mentioned.
Answer: These errors should have been rectified in the new version.

Reviewer 3 Report
The manuscript by Wilsens et al deals with the synthesis of a series of polyesters based on 1,6-hexanediol, sebacic acid, and N,N’-dimethylene-bis(pyrrolidone-4-carboxylic acid) (BP-C2). Copolymers with a varying amount of BP-C2 are synthesized in order to control the hydrolysis rate of the polymers in water or under bioactive conditions. The copolymers and their physicochemical properties were were characterized with, NMR, FT-IR, DSC, TGA, Lc-MS, waxs studies.
While the work looks interesting, experiments were done with care, there are a few key issues which need to be addressed before the manuscript can be recommended for publication:
The crystallinity of the resultant copolymer is a key issue and need further clarification as contradictory discussions are present in the manuscript. For example in line-269 the authors stated that "BP-C2 moieties hamper the crystallization of the poly(hexamethylene sebacate) segments, resulting in the formation of defected crystals with decreasing lamellar thickness." If this is true then same should be reflected in WAXS pattern. However, in line-279 authors stated that "These findings confirm that the BP-C2 moieties do not participate in the crystallization and thus reside in the amorphous phase. " If BP-C2 affects preferentially the amorphous phase of the polymer the crystalline domain will remain essentially unaffected. If this is the case, how the copolymerization would address the issue raised in line 47-48 in introduction? Authors need to add how a molecular weight of 20 kg/mol and higher would be beneficial.
I feel use of the term 'relative intensity' would be more appropriate while describing FT-IR.
I would suggest to perform DSC (with cooling rate not more than 5 degree C/min) for assessing the crystallization of bound water with the polymer.
Author Response
Before providing point-by-point answers to the queries raised by the reviewers, we would like to express our gratitude for their efforts in taking time to evaluate our work and provide critical comments with the aim to improve the quality of this work.
Reviewer 3
The manuscript by Wilsens et al deals with the synthesis of a series of polyesters based on 1,6-hexanediol, sebacic acid, and N,N’-dimethylene-bis(pyrrolidone-4-carboxylic acid) (BP-C2). Copolymers with a varying amount of BP-C2 are synthesized in order to control the hydrolysis rate of the polymers in water or under bioactive conditions. The copolymers and their physicochemical properties were were characterized with, NMR, FT-IR, DSC, TGA, LC-MS, waxs studies.
While the work looks interesting, experiments were done with care, there are a few key issues which need to be addressed before the manuscript can be recommended for publication:
The crystallinity of the resultant copolymer is a key issue and need further clarification as contradictory discussions are present in the manuscript. For example in line-269 the authors stated that "BP-C2 moieties hamper the crystallization of the poly(hexamethylene sebacate) segments, resulting in the formation of defected crystals with decreasing lamellar thickness." If this is true then same should be reflected in WAXS pattern.
Answer: The WAXS analysis indirectly confirms this as the crystal structure of the poly(hexamethylene sebacate) does not change with the incorporation of the BP-C2 in the polymer backbone. This can be detected by the fact that the peaks locations do not notably shift or change in width, but instead only decrease in intensity suggesting that only the overall crystallinity in the sample decreases. In general, when generating defected crystals (BP-C2 does not co-crystallize but “sit in the way”), the melting temperature decreases as a result of the decrease in lamellar thickness. To recall, the melting temperature of polymers having chain-folded crystals is determined by the lamellar thickness; the larger the lamella, the closer the observed melting temperature with be to the equilibrium melting temperature.
To prevent any confusion on this matter, we have changed the word “hamper” to “limit”.
However, in line-279 authors stated that "These findings confirm that the BP-C2 moieties do not participate in the crystallization and thus reside in the amorphous phase. " If BP-C2 affects preferentially the amorphous phase of the polymer the crystalline domain will remain essentially unaffected. If this is the case, how the copolymerization would address the issue raised in line 47-48 in introduction?
Answer: Indeed, this remark of the reviewer is important. As the BP-C2 does not affect the contents of the crystalline fraction (only the overall crystallinity), it will not affect the degradation of the poly(hexamethylene sebacate) crystals. To address this, we have added the following component in the conclusion of the work:
“Overall, the results in this work demonstrate that the introduction of the hydrophilic BP-C2 moiety is a viable route to enhance the rate of hydrolysis of hydrophobic polyesters in the presence of water and under bioactive conditions, though only of the amorphous component of the polymers.”
Authors need to add how a molecular weight of 20 kg/mol and higher would be beneficial.
Answer: We have added the following to the section to highlight that such values are common polymers synthesized via melt-polycondensation techniques.
“Nevertheless, the use of BP-C2 appear to limit the molecular weight build-up during polymerization as we obtain polymers with a molecular weight (Mw) close to 20 kg/mol or higher, values common for polymer synthesized via melt-polycondensation routes.”
I feel use of the term 'relative intensity' would be more appropriate while describing FT-IR.”
Answer: The FTIR spectra are shown as collected, with their absorbance on the y-axis. No shifting or normalization of the data was performed. For this reason, we prefer the use of absorbance.
I would suggest to perform DSC (with cooling rate not more than 5 degree C/min) for assessing the crystallization of bound water with the polymer.
Answer: The DSC data has been collected on the dry samples, hence there should not be any water present in the crystals (as is evident from the WAXD data). Furthermore, the crystallization in the presence of water is rather challenging: As is demonstrated in references 23 and 24, polymers containing bispyrrolidone based monomers are highly sensitive to absorb water, effectively facilitating a suppression in glass transition temperature. Such increased mobility is expected to alter the crystallization kinetics, furthermore, placing these low melting / amorphous polyesters in a DSC pan in the presence of water is undoubtedly resulting in rapid hydrolysis upon heating. For these reasons, we prefer to restrict ourselves to the identification of the melting behavior in the dry state.
Reviewer 4 Report
The manuscript by Wilsens and co-workers reported interesting studies on the incorporation of pyrrolidone units in polyesters and the impact on the water uptake and hydrolysis behaviors. The paper is overall well-written, the polymers with various content of BP-C2 were well characterized and studied, and the major conclusion was supported by the experimental results. The polyesters reported in this work can be very useful as renewable biomaterials with tunable degradation profiles. I suggest its acceptance for Polymers, but have some challenging questions for the authors. The authors may want to solve these questions in follow-up publications but it will be helpful if they can provide their opinions and plans.
The water adsorption and hydrolysis studies (Figs. 6 and 7, and related discussions) are interesting but may require further experiments to demonstrate the potential of the polyesters for real biomedical applications. Specifically,
(1) The authors may want to test more polymers with 40-70% BP-C2 contents to elucidate the abnormal enzymatic hydrolysis behavior of 50% BP-C2. If 75% BP-C2 and 100% BP-C2 are more like “surface erosion”, does 50% BP-C2 have a “bulk erosion” behavior? What is the composition window of the “bulk erosion” hydrolysis profiles for these polyesters?
(2) The different degradation profile can be more clearly demonstrated when incorporating some reporter molecules, like a fluorescent dye. It’s interesting to see how different the dye release profile is when tuning the contents of BP-C2.
(3) What is the impact of solution pH on the hydrolysis behavior of the polyesters?
Some minor issues:
(1) All the maintext figure citations appear to be errors. Please correct them.
(2) There are three “Section 3.1” in the manuscript.
Author Response
Before providing point-by-point answers to the queries raised by the reviewers, we would like to express our gratitude for their efforts in taking time to evaluate our work and provide critical comments with the aim to improve the quality of this work.
Reviewer 4
The manuscript by Wilsens and co-workers reported interesting studies on the incorporation of pyrrolidone units in polyesters and the impact on the water uptake and hydrolysis behaviors. The paper is overall well-written, the polymers with various content of BP-C2 were well characterized and studied, and the major conclusion was supported by the experimental results. The polyesters reported in this work can be very useful as renewable biomaterials with tunable degradation profiles. I suggest its acceptance for Polymers, but have some challenging questions for the authors. The authors may want to solve these questions in follow-up publications but it will be helpful if they can provide their opinions and plans.
The water adsorption and hydrolysis studies (Figs. 6 and 7, and related discussions) are interesting but may require further experiments to demonstrate the potential of the polyesters for real biomedical applications.
Answer: We are thankful for the kind words of the reviewer. Indeed, at the moment we are conducting cell toxicity studies of these series of polymers, in particular on the BPDA based monomers, as no information is available on their toxicity. Current results indicate that these BPDA based polymers are not toxic when brought in contact to cells, however, the effect of the hydrolysis and release of oligomers / monomers on cell viability is still ongoing.
Specifically,
(1) The authors may want to test more polymers with 40-70% BP-C2 contents to elucidate the abnormal enzymatic hydrolysis behavior of 50% BP-C2. If 75% BP-C2 and 100% BP-C2 are more like “surface erosion”, does 50% BP-C2 have a “bulk erosion” behavior? What is the composition window of the “bulk erosion” hydrolysis profiles for these polyesters?
- Answer: Indeed the reviewer in correct in stating that this particular region requires further evaluation. Preliminary SEM experiments have been performed, which suggest that the decrease in molecular weight might also be the result from the fragmentation of the sample, hence the generation of more surface. Nevertheless, this finding still supports the possibility of bulk erosion behavior and requires further evaluation.
(2) The different degradation profile can be more clearly demonstrated when incorporating some reporter molecules, like a fluorescent dye. It’s interesting to see how different the dye release profile is when tuning the contents of BP-C2.
- Answer: Indeed, this would be an interesting addition to the current experiments we are conducting on the BPDA based polymers in general. Recently, we reported on the thermal curing of bis(2-oxazoline) based thermosets, which readily react with these BPDA based monomers. It would be highly interesting to perform the degradation studies while chemically incorporating such fluorescent dyes in the polymer backbone and then monitor their release profile as hydrolysis occurs. Thank you for this suggestion.
(3) What is the impact of solution pH on the hydrolysis behavior of the polyesters?
- Answer: Generally, we notice that the pyrrolidone rings are slightly basic in nature, hence they dissolve better upon introduction in a slightly acidic medium. This is expected to significantly enhance the hydrolysis of the polymer chains. In contrast, they do not prefer basic media, which will limit the water absorption and likely the accessibility of the ester bonds for the enzyme. In addition, we obviously deal with the acid or base catalyze hydrolysis.
Some minor issues:
(1) All the maintext figure citations appear to be errors. Please correct them.
(2) There are three “Section 3.1” in the manuscript.
- Answer: Both queries should have been addressed in the new version.
Round 2
Reviewer 2 Report
Authors have tried to improve the most problematic point. Now, the experimental limitations caused by the nature of the studied polymers can be better understood. The paper can be published despite no additional effort concerning the experimental part has been performed.
Reviewer 3 Report
I'm mostly satisfied with the response from the authors, and therefore would recommend acceptance of the manuscript in its present form